# Exposure to House Dust Mite Allergen and Endotoxin in Early Life and Sensitization and Allergic Rhinitis: The JECS

**DOI:** 10.3390/ijerph192214796

**Published:** 2022-11-10

**Authors:** Reiji Kojima, Ryoji Shinohara, Megumi Kushima, Sayaka Horiuchi, Sanae Otawa, Kunio Miyake, Hiroshi Yokomichi, Yuka Akiyama, Tadao Ooka, Zentaro Yamagata

**Affiliations:** 1Department of Health Sciences, School of Medicine, University of Yamanashi, Shimokato, Chuo 1110, Kofu 409-3898, Yamanashi, Japan; 2Center for Birth Cohort Studies, University of Yamanashi, Shimokato, Chuo 1110, Kofu 409-3898, Yamanashi, Japan

**Keywords:** allergic rhinitis, endotoxin, house dust mite, sensitization

## Abstract

The association between endotoxin and allergic rhinitis (AR) is not conclusive. The aim of this study was to determine the association between endotoxin and house dust mite (HDM) allergens in dust, and HDM sensitization and AR among Japanese infants. This study included 4188 participants in the Sub-Cohort Study of the Japan Environment and Children’s Study. Dust was collected from children’s mattresses at age 18 months and endotoxin and HDM allergen levels were measured. A logistic regression model was used to analyze the association between endotoxin or HDM allergen and the sensitization to HDM (specific-IgE) at age 2 and AR at age 3. The median (interquartile range) endotoxin level was 375.1(186.9–826.5) EU/m^2^ and the Der 1 (Der p 1 + Der f 1) level was 51.2 (14.8–168.6) ng/m^2^. There were significant positive associations between endotoxin and HDM sensitization (Der f 1, adjusted odds ratio [aOR] quartile [Q] 4 vs. Q1, 1.44, 95% CI, 1.04–2.00; Der p 1, aOR Q4 vs. Q1, 1.56, 95% CI, 1.12–2.16). There were also significant positive associations between Der 1 exposure and HDM sensitization (Der f 1, aOR Q3 vs. Q1, 1.75, 95% CI, 1.26–2.44; aOR Q4 vs. Q1, 2.98, 95% CI, 2.15–4.13; Der p 1, aOR Q3 vs. Q1, 1.91, 95% CI, 1.37–2.66; aOR Q4 vs. Q1, 2.91, 95% CI, 2.09–4.05). There were, however, no associations between endotoxin or Der 1 and AR. In the population residing mostly in non-farming settings, both endotoxin and HDM allergens in dust were associated with an increased risk of HDM allergen sensitization, but not with AR.

## 1. Introduction

In 1989, Strachan presented the hygiene hypothesis, which proposes that reduced opportunities for infection and exposure to environmental bacteria and other antigens in infancy, caused by improved hygiene, promote the development of allergies [1]. Braun-Fahrlander et al. subsequently reported high environmental endotoxin levels on livestock farms and that endotoxin levels in the house were inversely correlated with subsequent sensitization and allergic rhinitis (AR) [2]. An endotoxin is a cell-wall component of Gram-negative bacteria. Research on Amish and Hutterite communities has also reported higher levels of endotoxin and lower frequencies of sensitization and asthma from dust in Amish households with traditional farming practices, and this has been verified in mouse models [3]. Mouse models have shown that endotoxin induce differentiation of naive T cells into Th1 cells via innate immunity, suppress allergic sensitization, and have a protective effect against the development of allergies [4]. Recently, studies of a mouse model reported that endotoxin stimulation of Toll-like receptor 4 (TLR-4) increases the expression of tumor necrosis factor alpha-induced protein 3 (TNFAIP3)/A20 in the airway epithelium and inhibits antigen uptake by dendritic cells [5].

The results of epidemiological studies on the association between endotoxin, sensitization and allergies have been inconsistent [6,7,8]. Cohort studies have shown that early childhood exposure to endotoxin is associated with increased wheezing [9], but results are inconsistent for the development of asthma [6,10]. endotoxin have, however, been reported to cause acute exacerbations in asthmatics [11]. Tischer et al. found reduced incidence of AR associated with infantile endotoxin exposure in a German cohort, but no significant association in a Dutch cohort [12]. A review by Simpson et al. [7] found that more than half of epidemiologic studies reported a preventive effect of endotoxin against allergic sensitization, while the remaining studies found no significant association, and one study reported that endotoxin were a risk for allergic sensitization [13]. Most studies of the association between environmental endotoxin and allergic diseases or sensitization, which required dust collection, have relatively small sample sizes [7]. In addition, although dose–response relationships between house dust mite (HDM) exposure and sensitization to HDM have been reported [14,15], studies analyzing HDM allergen and endotoxin simultaneously are limited [15].

The purpose of this study was to determine the association between early exposure to endotoxin and HDM allergens and HDM sensitization at age 2 years, and AR at age 3 years and in the general population.

## 2. Materials and Methods

### 2.1. Participants

The participants in this study were mother–child pairs who participated in the Japan Environment and Children’s Study (JECS) Sub-Cohort Study [16,17,18]. The protocol and baseline data of the JECS and the JECS Sub-Cohort Study have been described elsewhere [16,17,18]. Briefly, the JECS is an ongoing, nationwide, birth-cohort study, which recruited more than 100,000 pregnant women who lived in one of Study Areas covering a wide geographical area of Japan, from January 2011 to March 2014 [16,17]. The JECS Sub-Cohort Study, which includes face-to-face assessment of neuropsychiatric development, pediatric examination, blood and urine collection for clinical testing, and home visits (ambient and indoor air measurement and dust collection) was conducted with 5017 children randomly selected from 100,303 children in the JECS Main Study [18]. The JECS protocol was reviewed and approved by the Ministry of the Environment’s Institutional Review Board on Epidemiological Studies (IRB number: 100910001) and by the Ethics Committees of all participating institutions. Written informed consent was obtained from all participants.

This study was based on the JECS dataset jecs-ta-20190930, which was released in October 2019. The dataset of Sub-Cohort Study included 5017 records. We excluded participants with missing endotoxin and dust allergen data (*n* = 333), IgE data or AR information (*n* = 496). The final study population included 4188 mother–child pairs.

### 2.2. Variables

#### 2.2.1. Outcomes

The outcomes of this study were HDM allergen sensitization at 2 years old and AR (doctor-diagnosed as well as symptoms in the past year) at 3 years old. Sensitization to HDM allergen was determined by blood immunoglobulin E (IgE) levels specific to *Dermatophagoides farinae* 1 (Der f 1) and *Dermatophagoides pteronyssinus* 1 (Der p 1). HDM allergen-specific IgE levels in blood were measured with densely carboxylated protein microarrays and expressed as binding units per volume (BUe/mL), which correlate strongly with allergen-specific IgE values determined with the UniCAP system [19,20,21]. Although the cutoff value of specific-IgE in 2-year-old children is unclear, the prevalence of HDM sensitization in 2-year-old children has been reported to be at most 10% [22]. Sensitization to HDM was therefore defined as a specific-IgE level above the upper tenth percentile (Der f 1 ≥ 78.23 BUe/mL; Der p 1 ≥ 118.02 BUe/mL). Information about AR was reported by the mothers via self-administered questionnaires [23].

#### 2.2.2. Exposure

This study evaluated exposure to endotoxin and HDM allergen in dust from children’s mattresses at 18 months of age [18]. Dust was collected by trained staff using standardized methods. A 50 cm × 100 cm frame, exclusively for sampling, was placed on the mattress and dust in this area was collected with a vacuum cleaner (Model DC61, Dyson; Tokyo, Japan) for two minutes. The dust was frozen until analysis. endotoxin were measured with Kinetic Chromogenic LAL Assay (Kinetic-QCL; Lonza Japan; Tokyo, Japan). In accordance with previous studies [10,15], HDM allergens (Der p 1 and Der f 1) were measured with enzyme-linked immunosorbent assay (ELISA) kits (Indoor Biotechnologies Ltd.; Charlottesville, VA, USA). Dust samples with undetectable endotoxin, Der p 1 or Der f 1, were assigned a value of limit of detection/2 to calculate participants’ indoor exposure. Der p 1 and Der f 1 summed to Der 1. Exposures were expressed both as concentration (endotoxin or HDM allergen per gram of sampled dust) and load (endotoxin or HDM allergen per square meter of sampling surface area).

### 2.3. Statistical Analyses

We calculated summary measures of characteristics of the study participants, exposures, and outcomes. Logistic regression analyses between endotoxin or HDM allergen (Der 1) loads in dust and HDM sensitization and AR prevalence were performed to calculate odds ratios (OR) and 95% confidence intervals (CIs). Endotoxin or HDM allergen loads were divided into quartiles (Q) and the references for the analyses were assigned to the lower quartile, Q1. Model 1 was a crude model. For Model 2, we selected several covariates, based on previously reported pregnancy and early infancy factors related to the development of allergies in children [8] and directed acyclic graphs. We adjusted for parental allergy history, passive smoking, annual household income, mode of delivery, birth weight, sex, the presence of older siblings, exclusive breastfeeding, daycare attendance at one year, and household pets. To examine the modification of the effect of endotoxin by Der p1, subgroup analyses by level of Der 1 (cutoff point was median) were performed. To examine the linearity between endotoxin and Der 1 and HDM sensitization, an additional multivariate linear regression analysis was performed with variable log transformation. All statistical analyses were performed using SAS 9.4 (SAS Institute Inc., Cary, NC, USA). Statistical significance was considered as *p* < 0.05.

## 3. Results

Table 1 shows the study participant characteristics. Only 28 (0.7%) children were living in an environment with livestock, such as horses, cattle, pigs, and poultry. Table 2 shows endotoxin and Der 1 levels in dust samples. The median (interquartile range) endotoxin load was 375.1(186.9–826.5) EU/m^2^ and Der 1 load was 51.2 (14.8–168.6) ng/m^2^. The correlation coefficient between log_10_-transformed endotoxin load and Der 1 load was 0.43 (Pearson’s correlation coefficient, *p* < 0.01). The percentages of AR at 3 years old were 213 (5.1%) and 1303 (31.1%) for doctor-diagnosed and parent-reported symptoms, respectively. Table 3 presents the associations between endotoxin or HDM allergen exposure and HDM sensitization or AR. We observed significant positive associations between endotoxin and HDM sensitization (Der f 1, adjusted odds ratio [aOR] quartile [Q] 4 vs. Q1, 1.44, 95% CI, 1.04–2.00; Der p 1, aOR Q4 vs. Q1, 1.56, 95% CI, 1.12–2.16). There were significant positive associations between Der 1 exposure and HDM sensitization (Der f 1, aOR Q3 vs. Q1, 1.75, 95% CI, 1.26–2.44; aOR Q4 vs. Q1, 2.98, 95% CI, 2.15–4.13; Der p 1, aOR Q3 vs. Q1, 1.91, 95% CI, 1.37–2.66; aOR Q4 vs. Q1, 2.91, 95% CI, 2.09–4.05). There were no associations between endotoxin or Der p 1 exposure and AR (doctor diagnosed or symptom). Subgroup analysis by Der 1 level (cutoff: median, 51.2 ng/m^2^) showed that there was no modification of the effect of endotoxin on HDM sensitization by Der 1 level. (Table 4). Results from the multivariate linear regression analysis showed that endotoxin and Der 1 were both positively and linearly associated with HDM sensitization (Appendix A).

## 4. Discussion

This study investigated the relationship between endotoxin and HDM allergens in dust collected at 18 months and HDM sensitization at age 2 and AR at age 3. Both endotoxin and HDM allergen levels in dust were associated with an increased risk of HDM sensitization at age 2. However, neither endotoxin nor HDM allergens were associated with AR at age 3 years. The effect of endotoxin on HDM allergen sensitization was independent of HDM allergen levels.

Many studies have reported that exposure to endotoxin in infancy has a protective effect against allergic sensitization [2,7,24], although some studies have found no significant association [25,26]. The preventive effect is thought to depend on the endotoxin levels, timing of exposure, genetic predisposition, and allergy history [7]. In this study, exposure to higher endotoxin levels was associated with a significantly higher prevalence of HDM allergen sensitization. Although this result contradicts the previous report [7], the discrepancy might be due to differences in endotoxin dosage. In a study of farm families, Braun-Fahrlander et al. reported that higher endotoxin levels in a child’s mattress were associated with fewer allergies and sensitization [2]. The median endotoxin level was 29,897 U/m^2^, which is much higher than the median level of 375.1 U/m^2^ in our study. Furthermore, the majority of the participants in our study resided in urban settings. The amount of endotoxin in the present study was lower than in the farm-based study, and therefore the amount of endotoxin might be a risk for allergic sensitization. Other farm-based studies have found no significant association between endotoxin and sensitization when the endotoxin load was as low as 3000 U/m^2^ [26]. Studies in urban areas have shown a protective effect when the endotoxin load was about 3000 U/m^2^ [24], but no significant association was seen with a load of less than 1000 U/m^2^ [25]. In the only study which found endotoxin to be a risk for sensitization, endotoxin load was 2601 U/m^2^, but the concentration was as low as 8.8 U/mg [13]. The JECS [10], which used the same dataset as our study, reported higher endotoxin levels to be positively associated with early childhood asthma and total IgE. In mouse models, low levels of lipopolysaccharide/endotoxin have been reported to promote ovalbumin sensitization [27] and high levels of endotoxin to prevent ovalbumin sensitization [28]. In environments with low endotoxin levels, endotoxin might be a risk for sensitization to HDM allergens, and the relationship between endotoxin and sensitization might be bell-shaped. To find the threshold, studies in populations with a wider range of endotoxin levels are warranted.

The relationship between endotoxin and AR has been inconclusive in previous studies [2,12]. In our study, there was no association between the level of endotoxin and the development of AR. We investigated AR at age 3 years, which might have skewed the results as AR onset could occur after the age of 3 years. Follow-up studies are therefore needed. In our study, HDM allergens in dust were significantly associated with HDM sensitization. This result was consistent with the results of previous studies [14,15]. However, the effect of endotoxin on HDM sensitization was independent of the HDM allergen levels. Previous studies have reported a bell-shaped association between HDM allergens and sensitization, with higher endotoxin levels leading to lower peaks [15]. It is possible that the range of endotoxin levels in our study was insufficient for observing this effect.

Strengths of this study include the relatively large sample size from the general population [17], standardized dust collection and assessment of endotoxin and allergens [18], and the simultaneous analysis of Der 1 and endotoxin. Limitations of the study include the self-reported AR assessment and the one-time dust collection. However, endotoxin levels were reported to be relatively stable and correlated well with re-measurements [29]. Regarding HDM allergen, a previous study reported high correlations of Der 1 levels in re-measurements [15]. However, because of the seasonal variation of HDM allergen, we performed an analysis stratified by dust collection season (Appendix A). As a result, there was almost no difference by collection season. According to Ege et al., exposure to environments with high bacterial diversity has a protective effect against the onset of sensitization [30]. Although the present study focused on endotoxin levels, future analyses should also take bacterial diversity into account. The present study did not consider the genetic variability of the participants. endotoxin are recognized by the innate immune system, which has the trimolecular complex of CD14/TLR4/MD2 at the core [7]. An interaction between genetic polymorphisms of CD14 and endotoxin exposure has been reported [7], and studies that take genetic variation into account are warranted.

## 5. Conclusions

In the population residing mostly in non-farming settings, the level of endotoxin in dust was associated with an increased risk of HDM allergen sensitization. HDM allergen levels were also associated with an increased risk of HDM sensitization. However, neither endotoxin nor HDM allergen in dust was associated with AR at age 3 years. As the prevalence of AR increases after school age, follow-up studies are needed.

## Figures and Tables

**Table 1 ijerph-19-14796-t001:** Characteristics of the study participants (n = 4188).

		Number	(%)
Maternal allergy history			
	Yes	2129	(50.8)
	No	2059	(49.2)
Paternal allergy history			
	Yes	960	(22.9)
	No	1273	(30.4)
	Unknown	1955	(46.7)
Maternal age at pregnancy (years)			
	<25	411	(9.8)
	25–30	1080	(25.8)
	30–35	1504	(35.9)
	≥35	1193	(28.5)
Household income (million JPY/year)			
	< 2	176	(4.2)
	2–4	1274	(30.4)
	4–6	1383	(33.0)
	≥ 6	1166	(27.8)
	Unknown	189	(4.5)
Mode of delivery			
	Caesarean section	764	(18.3)
	Vaginal	3416	(81.7)
Birth weight (g)			
	<2500	338	(8.1)
	≥2500	3850	(91.9)
Child’s sex			
	Male	2138	(51.1)
	Female	2050	(49.0)
Older siblings			
	Yes	2348	(56.1)
	No	1840	(43.9)
Duration of exclusive breastfeeding			
	<6 months	2391	(57.5)
	≥6 months	1770	(42.5)
Daycare attendance at 1 year old			
	Yes	1121	(27.1)
	No	3022	(72.9)
Household pet at 18 months			
	Yes	724	(17.5)
	No	3422	(82.5)
Any domestic animals at 6 months			
	Yes	28	(0.7)
	No	4160	(99.3)
Passive smoking at 18 months			
	None	3250	(78.7)
	Sometimes	740	(17.9)
	Frequently	141	(3.4)

**Table 2 ijerph-19-14796-t002:** Endotoxin and Der 1 levels in dust samples.

	Median (IQR) or N (%)
Endotoxin (EU/mg)	17.0 (11.0–30.0)
Endotoxin (EU/m^2^)	375.1 (186.9–826.5)
Endotoxin (BLD)	0 (0.0)
Der f 1 (BLD)	1920 (45.9)
Der p 1 (BLD)	2055 (49.1)
Der 1 (μg/g)	2.4 (0.8–7.1)
Der 1 (ng/m^2^)	51.2 (14.8–168.6)
Dust (mg)	9.8 (7.8–16.8)

Abbreviations: BLD, below limit of detection, Der f 1, *Dermatophagoides farinae* 1, Der p 1, *Dermatophagoides pteronyssinus* 1, Der p 1 and Der f 1 summed to Der 1; IQR, inter-quartile range.

**Table 3 ijerph-19-14796-t003:** Odds ratios of sensitization to Der 1 and allergic rhinitis in relation to endotoxin or Der 1 exposure in dust.

			Der f 1 Sensitization			Der p 1 Sensitization			Doctor Diagnosed Allergic Rhinitis			Nasal Symptoms		
			OR	95% CI		OR	95% CI		OR	95% CI		OR	95% CI	
Endotoxin (EU/m^2^)														
	Model 1													
	Q1	<186.9	ref			ref			ref			ref		
	Q2	≥186.9, <375.1	**1.41**	**1.03**	**1.94**	**1.51**	**1.10**	**2.08**	1.00	0.68	1.47	1.14	0.95	1.37
	Q3	≥375.1, <826.5	**1.61**	**1.18**	**2.19**	**1.61**	**1.18**	**2.21**	1.06	0.72	1.55	1.05	0.87	1.26
	Q4	≥826.5	**2.06**	**1.53**	**2.77**	**2.22**	**1.65**	**3.01**	0.88	0.59	1.32	1.06	0.88	1.27
	Model 2													
	Q1	<186.9	ref			ref			ref			ref		
	Q2	≥186.9, <375.1	1.16	0.84	1.62	1.23	0.88	1.71	0.96	0.64	1.43	1.10	0.91	1.34
	Q3	≥375.1, <826.5	1.19	0.85	1.65	1.16	0.83	1.63	0.98	0.65	1.48	0.98	0.80	1.20
	Q4	≥826.5	**1.44**	**1.04**	**2.00**	**1.56**	**1.12**	**2.16**	0.80	0.51	1.23	1.02	0.83	1.25
Der 1 (ng/m^2^)														
	Model 1													
	Q1	<14.8	ref			ref			ref			ref		
	Q2	≥14.8, <51.2	1.11	0.79	1.57	1.13	0.80	1.61	0.95	0.63	1.44	0.92	0.76	1.11
	Q3	≥51.2, <168.6	**1.79**	**1.30**	**2.46**	**2.00**	**1.45**	**2.75**	1.26	0.86	1.85	1.02	0.85	1.23
	Q4	≥168.6	**2.96**	**2.20**	**4.00**	**3.07**	**2.27**	**4.17**	1.15	0.78	1.70	1.05	0.87	1.26
	Model 2													
	Q1	<14.8	ref			ref			ref			ref		
	Q2	≥14.8, <51.2	1.09	0.77	1.56	1.11	0.77	1.59	0.97	0.64	1.48	0.91	0.75	1.10
	Q3	≥51.2, <168.6	**1.75**	**1.26**	**2.44**	**1.91**	**1.37**	**2.66**	1.32	0.88	1.98	1.01	0.83	1.23
	Q4	≥168.6	**2.98**	**2.15**	**4.13**	**2.91**	**2.09**	**4.05**	1.21	0.78	1.88	1.04	0.84	1.27

Abbreviations: CI, confidence interval; OR, odds ratio. Model 1: crude; Model 2: Adjusted for parental allergy history, passive smoking, household income, mode of delivery, birth weight, child’s sex, older siblings, exclusive breastfeeding, daycare attendance at 1 year, and household pet. Boldface indicates statistical significance (*p* < 0.05).

**Table 4 ijerph-19-14796-t004:** Odds ratios of sensitization to Der 1 and allergic rhinitis in relation to endotoxin exposure in the Der 1 status subgroup.

		Der f 1 Sensitization			Der p 1 Sensitization			Doctor-Diagnosed Allergic Rhinitis			Nasal Symptoms		
		aOR	95% CI		aOR	95% CI		aOR	95% CI		aOR	95% CI	
Endotoxin (EU/m^2^)													
Der 1 < 51.2 ng/m^2^													
Q1	<186.9	ref			ref			ref			ref		
Q2	≥186.9, <375.1	1.15	0.71	1.86	1.37	0.83	2.26	1.03	0.60	1.75	1.20	0.94	1.53
Q3	≥375.1, <826.5	1.50	0.93	2.42	**2.02**	**1.24**	**3.31**	1.20	0.69	2.11	1.04	0.79	1.36
Q4	≥826.5	1.61	0.98	2.67	**1.99**	**1.18**	**3.35**	0.65	0.32	1.28	0.90	0.67	1.20
Der 1 ≥51.2 ng/m^2^													
Q1	<186.9	ref			ref			ref			ref		
Q2	≥186.9, <375.1	1.14	0.72	1.81	1.04	0.67	1.62	0.82	0.44	1.51	0.93	0.67	1.29
Q3	≥375.1, <826.5	1.10	0.70	1.74	0.82	0.52	1.28	0.80	0.44	1.46	0.90	0.65	1.24
Q4	≥826.5	1.54	1.00	2.38	1.37	0.90	2.08	0.75	0.41	1.36	1.01	0.74	1.38

Abbreviations: CI, confidence interval; OR, odds ratio. Model 1: crude; Model 2: Adjusted for parental allergy history, passive smoking, household income, mode of delivery, birth weight, child’s sex, older siblings, exclusive breastfeeding, daycare attendance at 1 year, and household pet. Boldface indicates statistical significance (*p* < 0.05).

## Data Availability

Data are unsuitable for public deposition due to ethical restrictions and legal framework of Japan. It is prohibited by the Act on the Protection of Personal Information (Act No. 57 of 30 May 2003, amendment on 9 September 2015) to publicly deposit the data containing personal information. Ethical Guidelines for Medical and Health Research Involving Human Subjects enforced by the Japan Ministry of Education, Culture, Sports, Science and Technology and the Ministry of Health, Labour and Welfare also restricts the open sharing of the epidemiologic data. All inquiries about access to data should be sent to: jecs-en@nies.go.jp. The person responsible for handling enquiries sent to this e-mail address is Shoji F. Nakayama, JECS Programme Office, National Institute for Environmental Studies.

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
