# Peer review of "Exposure to House Dust Mite Allergen and Endotoxin in Early Life and Sensitization and Allergic Rhinitis: The JECS"

_ijerph, 2022, doi:10.3390/ijerph192214796_

Round 1

Reviewer 1 Report

In this study, the association between endotoxin and house dust mite (HDM) allergen in dust, and HDM 13 sensitization and allergic rhinitis (AR) among Japanese infants were investigated. I have major comments on the novelty and methodology of this study. 

1.      There are many studies focusing on the relationship between AR and endotoxin / HDM exposure. The uniqueness of this study is to combine the endotoxin and HDM exposure in a single study. However, both parameters could not be associated with AR. This is contradictory to previous findings and the possibility of methodological problem could not be ruled out. For example, the statistical model used to determine the crosstalk between endotoxin and HDM allergen is too simple and primitive.

2.      The study started with the determination of HDM allergens and endotoxin in 18 months and ended with AR assessment in age 3. In fact, a single end point of AR at age 3 is not sufficient and longer follow up may be necessary to discover the association.    

3.      Finally, the depth of analysis is limited for such a big dataset. Moreover, more statistical parameters besides the odd ratios should be employed to mine the correlation and statiscal significance among HDM allergens, endotoxin and AR. 

Reviewer 2 Report

I have reviewed the manuscript ‘Exposure to House Dust Mite Allergen and Endotoxin in Early 2 Life and Sensitization and Allergic Rhinitis: The JECS’ by Kojima and co-workers with much interest. The authors aims to detect an association between early exposure to endotoxin and HDM and AR later in life in a subchort of a larger environmental study, but unfortunately this association was not found. I think the most interesting hypothesis from this study is: In environments with low endotoxin levels, endotoxin might be a risk for sensitization to HDM allergens, and the relationship between endotoxin and sensitization might be bell-shaped. To find the threshold, studies in populations with a wider range of endotoxin levels are warranted.

Specific comments:

Abstract

Line 12-13: ‘To determine the association between endotoxin and house dust mite (HDM) allergen in dust, and HDM sensitization and AR among Japanese infants. ‘This sentence is not complete, how is this determined?

Line 19: ‘Der 1’ – is this Der p 1, Der f 1 or the total of both? Needs explanation

Introduction

Line 33: preferred ‘caused by’ instead of ‘owing to’

Line 35: might be good to explain in the introduction what endotoxin(s) are

Line 51-52: use ‘a German cohort’ and ‘a Dutch cohort’ instead of ‘the’ cohort

Line: 61-63: the purpose of the study should be better explained, especially the sentence part ‘collected from dust at 18 months of age’ is not sufficient to understand how the study was done and to what purpose.

Materials and methods

2.1 Participants, line 74 – delete ‘being’ (is being conducted)

Line 74: 5,017 children from the main study – how are these selected?

2.2.1 Outcomes, Line 87-88: why are only Der p 1 and Der f 1 measured as there are many other HDM allergens?

2.2.2 Exposure, Line 99-100: Dust was collected by trained staff using standardized methods. Are these described in literature? If not, how are these methods standardized?

Line 103: Again, why only focus on Der p1 and Der f 1?

Line 105-106: ‘half the below the detection limits ’is not very clear, better ‘are assigned a value of limit of detection/2’

Results

Line 126-127: ‘only 28 children had any livestock’ is a bit strange, Consider ‘were living in an environment with livestock’ or ‘were in close contact to livestock’. Another question, did you perform specific analysis on this outcome? Since they were not part of the adjustment?

Table 1: I think it is also interesting that very little children were in close contact to household pets. I know household pets were part of the adjustment, but I think it would be interesting to perform specific analysis on livestock and household pets. This could have added value to this paper since the overall outcome was no association of HDM/endotoxin exposure and AR at age 3. But what about e.g. cat exposure and AR since cat exposure is correlated to asthma later in life.

Discussion

Line 157: It says ‘many studies have reported…..’ but only 1 reference is given. Please either use the references of these studies, or clarify that these studies are described in Simpson et al.

Line 167: Could you explain the differences in measurement systems

Line 197:  ‘However, endotoxin levels were reported to be relatively stable and correlate well with re-measurements’. What about HDM?

Round 2

Reviewer 1 Report

The content of this article is much improved after the incorporation of new results.